# Association between *PD-1* and *PD-L1* Polymorphisms and the Risk of Cancer: A Meta-Analysis of Case-Control Studies

**DOI:** 10.3390/cancers11081150

**Published:** 2019-08-10

**Authors:** Mohammad Hashemi, Shima Karami, Sahel Sarabandi, Abdolkarim Moazeni-Roodi, Andrzej Małecki, Saeid Ghavami, Emilia Wiechec

**Affiliations:** 1Genetics of Non-communicable Disease Research Center, Zahedan University of Medical Sciences, Zahedan 9816743463, Iran; 2Department of Clinical Biochemistry, School of Medicine, Zahedan University of Medical Sciences, Zahedan 9816743175, Iran; 3Department of Clinical Biochemistry, Iranshahr University of Medical Sciences, Iranshahr 9916643535, Iran; 4Instititute of Physiotherapy and Health Sciences, The Jerzy Kukuczka Academy of Physical Education in Katowice, 40-065 Katowice, Poland; 5Department of Human Anatomy and Cell Science, Max Rady College of Medicine, Rady Faculty of Health Sciences, University of Manitoba, Winnipeg, MB R3E 0J9, Canada; 6Research Institute in Oncology and Hematology, CancerCare Manitoba, University of Manitoba, Winnipeg, MB R3E 3P5, Canada; 7Department of Clinical and Experimental Medicine, Linköping University, 58183 Linköping, Sweden

**Keywords:** apoptosis, *PD-1*, *PD-L1*, polymorphism, cancer, meta-analysis

## Abstract

A number of case-control studies regarding the association of the polymorphisms in the programmed cell death 1 (PD-1) and programmed cell death ligand 1 (PD-L1) genes with the risk of cancer have yielded inconsistent findings. Therefore, we have conducted a comprehensive, updated meta-analysis study to identify the impact of *PD-1* and *PD-L1* polymorphisms on overall cancer susceptibility. The findings revealed that *PD-1* rs2227981 and rs11568821 polymorphisms significantly decreased the overall cancer risk (Odds Ratio (OR) = 0.82, 95% CI = 0.68–0.99, *p* = 0.04, TT vs. CT+CC; OR = 0.79, 95% CI = 0.67–0.94, *p* = 0.006, AG vs. GG, and OR = 0.82, 95% CI = 0.70–0.96, *p* = 0.020, AG+AA vs. GG, respectively), while *PD-1* rs7421861 polymorphism significantly increased the risk of developing cancer (OR = 1.16, 95% CI = 1.02–1.33, *p* = 0.03, CT vs. TT). The *PD-L1* rs4143815 variant significantly decreased the risk of cancer in homozygous (OR = 0.62, 95% CI = 0.41–0.94, *p* = 0.02), dominant (OR = 0.70, 95% CI = 0.50–0.97, *p* = 0.03), recessive (OR = 0.76, 95% CI = 0.60–0.96, *p* = 0.02), and allele (OR = 0.78, 95% CI = 0.63–0.96, *p* = 0.02) genetic models. No significant association between rs2227982, rs36084323, rs10204525, and rs2890658 polymorphisms and overall cancer risk has been found. In conclusions, the results of this meta-analysis have revealed an association between *PD-1* rs2227981, rs11568821, rs7421861, as well as *PD-L1* rs4143815 polymorphisms and overall cancer susceptibility.

## 1. Introduction

Cancer, a main public health issue is the leading cause of death globally. It was estimated that there will be about 18.1 million new cases of cancer and 9.6 million cancer deaths in 2018 [1]. Thus, the etiology and pathogenesis of cancer has not been elucidated completely and their understanding is decisive. Genome-wide association studies (GWAS) have simplified the search for potential genetic variants that are implicated in many diseases including cancer and single nucleotide polymorphisms (SNPs) are well studied genetic variations found in human genome. The number of SNPs that have so far been identified to play an important role in cancer susceptibility is significant [2]. It has been proposed that the immune system plays a key role in resisting and eliminating cancer cells and can affect cancer susceptibility. One of the main hallmarks of cancer cells is the immune suppression and evasion [3]. 

Tumor cells express the programmed death-1-ligand 1 (PD-L1) as an adaptive, resistant mechanism to suppress the inhibitory receptor, namely programmed cell death-1 (PD-1) in order to evade host immunosurveillance [4]. PD-1, also known as PD1 and CD279, is a cell surface immunosuppressive receptor belonging to immunoglobulin superfamily and is encoded by the *PDCD1* gene [5,6,7]. PD-1, is a negative regulator of the immune system and is expressed on CD4+ T cells, CD8+ T cells, NKT cells, B cells, and monocytes [8,9]. The antitumor CD8+ T cells exhibit preferential expression of PD-1 leading to their exhaustion and functional impairment, which in turns lead to attenuated tumor-specific immunity disseminating tumor progression [10,11]. The PD-1 blockade elevates the magnitude of T cell response such as proliferation of T cells and production of effector cytokines [12]. Additionally, PD-L1 signaling through conserved sequence motifs confers resistance of cancer cells towards proapoptotic interferon (IFN)-mediated cytotoxicity [13]. 

PD-1/PD-L1 axis is an important pathway to maintain immune tolerance and prevent autoimmune diseases in the evolution of immunity [14,15,16]. Furthermore, it influences the balance between tumor immune surveillance and immune resistance [17,18]. Elevated PD-L1 expression in tumor cells or tumor-infiltrating lymphocytes (TILs) leads to the exhaustion of T cells [19], and hence attenuated tumor-specific immunity disseminating tumor progression [20]. Gene polymorphisms might affect the normal process of gene activation and transcriptional initiation, hence influence the quantity of mRNA and encoded protein [21]. Both *PD-1* and *PD-L1* are polymorphic. Several studies investigated the association between genetic polymorphisms of *PD-1* and *PD-L1* genes and the risk of various cancers, but the finding are still inconclusive [5,6,7,22,23,24,25,26,27,28,29,30,31,32,33,34,35,36,37,38,39,40,41,42,43,44,45,46,47,48,49,50,51,52]. Thus, we performed a comprehensive meta-analysis in order to study the association of polymorphisms in *PD-1* (rs2227981, rs2227982, rs11568821, rs7421861, rs36084323, and rs10204525) and *PD-L1* (rs4143815, and rs2890658) with the risk of cancer. The locations and base pair positions of single nucleotide polymorphisms (SNPs) in *PD-1* and *PD-L1* genes are presented in Table 1.

## 2. Results

### 2.1. Study Characteristics

A flow diagram of the study selection process is shown in Figure 1. For *PD-1* polymorphisms, 54 case-control studies from a total of 26 articles [5,6,7,22,23,24,25,26,27,28,29,30,31,32,33,34,35,36,37,38,39,40,41,42,43,52] examining the associations of 6 widely studied polymorphisms in *PD-1* gene and cancer risk were included in this meta-analysis. There were 16 studies involving 5622 cases and 5450 controls that reported the association between *PD-1* rs2227981 polymorphism and cancer. Eleven studies including 4766 cases and 5839 controls investigated the relationship between *PD-1* rs2227982 polymorphism and cancer. Nine studies with 1846 cases and 1907 cases reported the association between *PD-1* rs11568821 variant and cancer risk. Seven studies including 3576 cancer cases and 5277 controls studied the correlation between *PD-1* rs7421861 polymorphism and cancer. Seven studies involving 3589 cases and 4314 controls examined the association between *PD-1* rs36084323 polymorphism and cancer risk. Six studies including 3366 cancer cases and 4391 controls studied the relationship between *PD-1* rs10204525 polymorphism and cancer. 

For *PD-L1* polymorphisms, 13 case-control studies from 10 articles [27,38,44,45,46,47,48,49,50,51] that assessed the impact of two polymorphisms of *PD-L1* were included in the pooled analysis. Eight studies including 3030 cases and 4145 controls evaluated the association between *PD-L1* rs4143815 polymorphism and cancer risk. Five studies with 1909 cases and 1970 controls assessed the correlation between *PD-L1* rs2890658 variant and cancer risk. The characteristics of all these studies are shown in Table 2.

### 2.2. Main Analysis Results

#### 2.2.1. Association of *PD-1* Polymorphisms with Cancer Risk

The pooled analysis involving *PD-1* rs2227981 polymorphism revealed that this variant significantly decreased the overall cancer risk in recessive (OR = 0.82, 95% CI = 0.68–0.99, *p* = 0.04, TT vs. CT+CC) genetic models (Table 3 and Figure 2). 

In regard to *PD-1* rs11568821 polymorphism, the findings indicated that this variant significantly decreased the overall cancer risk in heterozygous (OR = 0.79, 95% CI = 0.67–0.94, *p* = 0.006, AG vs. GG) and dominant (OR = 0.82, 95% CI = 0.70–0.96, *p* = 0.020, AG+AA vs. GG) genetic models (Table 3).

The pooled analysis proposed that *PD-1* rs7421861 polymorphism significantly increased the risk of overall cancer in heterozygous (OR = 1.16, 95% CI = 1.02–1.33, *p* = 0.03, CT vs. TT) genetic models (Table 3).

No significant association was found between *PD-1* rs2227982, rs36084323, and rs10204525 polymorphisms and cancer susceptibility (Table 3).

We performed stratified analyses and the findings are summarized in Table 4. We observed that *PD-1* rs2227981 significantly decreased the risk of gastrointestinal (GI) cancer (OR = 0.68, 95% CI = 0.56–0.84, *p* = 0.000, TT vs. CC; OR = 0.60, 95% CI = 0.40–0.89, *p* = 0.011, TT vs. CT+CC; OR = 0.83, 95% CI = 0.75–0.91, *p* = 0.000, T vs. C), lung cancer (OR = 0.65, 95% CI = 0.44–0.97, *p* = 0.030, TT vs. CC; OR = 0.84, 95% CI = 0.71–0.99, *p* = 0.043, CT+TT vs. CC; OR = 0.83, 95% CI = 0.72–0.95, *p* = 0.009, T vs. C), and breast cancer (OR = 0.82, 95% CI = 0.70–0.06, *p* = 0.012, T vs. C). 

Furthermore, we found that the *PD-1* rs2227982 was associated with an increased risk of cancer in hospital based studies (OR = 1.22, 95% CI = 1.06–1.40, *p* = 0.006, CT vs. CC; OR = 1.20, 95% CI = 1.05–1.37, *p* = 0.008, CT+TT vs. CC). We also found a negative correlation between the *PD-1* rs2227982 polymorphism and the risk of gastrointestinal cancer (OR = 1.18, 95% CI = 1.04–1.34, *p* = 0.011, CT vs. CC; OR = 1.16 (95% CI = 1.03–1.30, *p* = 0.017, CT+TT vs. CC) and breast cancer risk (OR = 0.73, 95% CI = 0.59–0.90, *p* = 0.004, CT vs. CC; OR = 0.73, 95% CI = 0.57–0.93, *p* = 0.010, TT vs. CC; OR = 73, 95% CI = 0.60–0.89, *p* = 0.002, CT+TT vs. CC; OR = 0.85, 95% CI = 76–0.96, *p* = 0.010, T vs. C). With reference to the *PD-1* rs7421861, our finding proposed that this variant significantly increased the risk of cancer in hospital based studies (OR = 1.89, 95% CI = 1.01–1.40, *p* = 0.042, CT vs. TT) as well as gastrointestinal cancer (OR = 1.19, 95% CI = 1.01–1.40, *p* = 0.042, CT vs. CC). Moreover, a significantly reduce cancer risk in population-based studies (OR = 0.80, 95% CI = 0.66–0.97, *p* = 0.020, AG vs. GG) was observed regarding *PD-1* rs11568821 variant. The *PD-1* rs36084323 variant was however associated with an increased risk of cancer in hospital-based studies (OR = 1.17, 95% CI = 1.01–1.35, *p* = 0.042, AG+AA vs. GG). 

#### 2.2.2. *PD-L1* Polymorphisms and Cancer Risk

The pooled ORs results for the relationship between the *PD-L1* rs4143815 and rs2890658 polymorphisms and the risk of cancer are shown in Table 3. The *PD-L1* rs4143815 variant significantly decreased the risk of cancer in homozygous (OR = 0.62, 95% CI = 0.41–0.94, *p* = 0.02), dominant (OR = 0.70, 95% CI = 0.50–0.97, *p* = 0.03), recessive (OR = 0.76, 95% CI = 0.60–0.96, *p* = 0.02), and allele (OR = 0.78, 95% CI = 0.63–0.96, *p* = 0.02) genetic models (Table 3 and Figure 3). The pooled analysis did not support an association between *PD-L1* rs2890658 polymorphism and risk of cancer susceptibility (Table 3).

We did stratified analysis (Table 4) and the findings revealed that *PD-L1* rs4143815 polymorphism significantly reduced the risk of gastrointestinal cancer (OR = 0.68, 95% CI = 0.48–0.97, *p* = 0.032, CC vs. GG; OR = 0.59, 95% CI = 0.37–0.96, *p* = 0.033, CC vs. GG; OR = 0.64, 95% CI = 0.43–0.95, *p* = 0.028, CG+CC vs. GG; OR = 0.76, 95% CI = 0.59–0.98, *p* = 0.034, C vs. G) and hospital-based studies (OR = 0.75, 95% CI = 0.58–0.97, *p* = 0.030, CC vs. CG+GG; OR = 0.76, 95% CI = 0.58–0.99, *p* = 0.043, C vs. G). In regard to *PD-L1* rs2890658, a positive correlation between this variant and the risk of lung cancer (OR = 1.74, 95% CI = 1.37–2.19, *p* = 0.000, AC vs. AA; OR = 1.77, 95% CI = 1.41–2.23, *p* = 0.000, AC+CC vs. AA; OR = 1.72, 95% CI = 1.39–2.13, *p* = 0.000 C vs. A) was observed (Table 4).

### 2.3. Heterogeneity

As shown in Table 3, heterogeneity between the studies regarding the *PD-1* rs2227981, *PD-1* rs36084323, *PD-1* rs10204525, and *PD-L1* rs4143815 was observed in all genetic models. For *PD-1* rs2227982 polymorphism, our results showed no evidence of heterogeneity in the recessive model (TT vs. CT+CC). Regarding *PD-1* rs11568821, heterogeneity was not observed in the heterozygous, homozygous, dominant, and recessive genetic models. Similarly, no evidence of heterogeneity in the heterozygous, homozygous, and recessive genetic models of *PD-1* rs7421861 was found. Heterogeneity was not detected in the homozygous and recessive genetic models of the *PD-L1* rs2890658. 

### 2.4. Publication Bias

The potential publication bias of the studies included in the present meta-analysis was examined by Begg’s funnel plot and Egger’s test. The results of publication bias are summarized in Table 3. Based on the above analysis, no publication bias for the association of *PD-1* rs2227982, *PD-1* rs7421861, and *PD-L1* rs4143815 variants in all genetic models and cancer risk was demonstrated (Table 3 and Figure 4). 

As presented in Table 3 and Figure 5, no publication bias was observed in recessive genetic model of *PD-1* rs2227981. Obvious publication bias was not found in the heterozygous, dominant, and allele genetic models of the *PD-1* rs11568821 *and PD-L1* rs2890658 (Table 3). Moreover, the publication bias was not observed in heterozygous, dominant, recessive, and allele genetic models of the *PD-1* rs36084323 and *PD-1* rs10204525. (Table 3). 

### 2.5. Sensitivity Analysis

Sensitivity analysis was conducted by replicating analysis after neglecting one study at a time to estimate the effect of quality of studies on the final findings. Taken together, our findings from the meta-analysis of the correlation between analyzed polymorphisms and cancer susceptibility remained unchanged in the heterozygous (*PD-1* rs2227982, *PD-1* rs36084323 and *PD-1* rs10204525), homozygous (*PD-1* rs2227982, *PD-1* rs7421861, *PD-1* rs36084323, *PD-1* rs10204525 and *PD-L1* rs2890658), dominant (*PD-1* rs36084323 and *PD-1* rs10204525), recessive (*PD-1* rs2227982, *PD-1* rs7421861, *PD-1* rs36084323 and *PD-L1* rs2890658), and allele (*PD-1* rs2227982, *PD-1* rs7421861 *PD-1* rs10204525 and *PD-L1* rs2890658) genetic models (Figure 6). In regard to *PD-L1* rs4143815, the findings changed in the heterozygous, homozygous, dominant, recessive, and allele genetics models (Figure 7). 

## 3. Discussion

It has been proposed that environmental and genetic factors contribute to cancer development [53,54]. Single nucleotide polymorphisms (SNPs) can be considered as biological markers that help scientists to recognize genes that are related to cancer [55]. 

PD-1 and PD-L1 are involved in the regulation of programmed cell death, which is the regulator of cancer cell proliferation as well as primary response in many cancer therapy strategies. Several studies have investigated the association between *PD-1* as well as *PD-L1* polymorphisms and the risk of various types of cancers; however, the findings remain discrepant. This meta-analysis provides, for the first time a quantitative estimated of the association between six SNPs of *PD-1* and two SNPs of *PD-L1* gene and cancer susceptibility. The findings indicated that *PD-1* rs2227981 and rs11568821 polymorphisms as well as *PDL-1* rs4143815 variant significantly decreased the overall cancer risk, while *PD-1* rs7421861 polymorphism significantly increased the risk of overall cancer. Our findings revealed no significant association between *PD-1* rs2227982, *PD-1* rs36084323, PD-1 rs10204525, and *PD-L1* rs2890658 polymorphisms and overall cancer risk.

We performed stratified analyses and our findings indicate that *PD-1* rs2227981 significantly decreased the risk of gastrointestinal cancer, lung cancer and breast cancer. The *PD-1* rs2227982 was associated with increased risk of cancer in hospital-based studies and lower risk of gastrointestinal and breast cancer. Similarly to *PD-1* rs7421861, the *PD-1* rs7421861 and *PD-1* rs36084323 variants significantly increased the risk of cancer in hospital-based studies. The *PD-1* rs11568821 was linked to reduce risk of cancer in population-based studies. Moreover, our findings revealed that *PD-L1* rs4143815 polymorphism significantly reduced the risk of gastrointestinal cancer and hospital-based studies. A positive correlation between *PD-L1* rs2890658 variant and the risk of lung cancer was observed.

Recently, Zou et al. [56] performed a meta-analysis of the association between *PD-L1* rs4143815 polymorphism and the risk of cancer and found also a significant association between this variant and cancer risk, which is in line with our findings. Like our results, a meta-analysis conducted by Da et al. [57] revealed no significant association between *PD-1* rs36084323 polymorphism and overall cancer susceptibility. Similar to previous meta-analysis conducted by Zhang et al. [58], we have also found that *PD-1* rs2227981 and rs11568821 polymorphisms were associated with decreased cancer susceptibility. In another study, Dong et al. [59] conducted a meta-analysis aimed to inspect the associations between *PD-1* rs2227981, rs2227982, rs7421861, and rs11568821 polymorphisms and cancer risk. There were seven studies involving 3395 cases and 2912 controls for *PD-1* rs2227981, four studies including 1961 cases and 2390 controls for *PD-1* rs2227982, four studies with 1975 cases and 2403 controls for *PD-1* rs7421861, and four studies for PD-1 rs11568821 variant and cancer risk. They have found that rs2227981 and rs11568821 polymorphisms significantly decreased the risk of cancer. Mamat et al. [60] conducted a meta-analysis of six studies involving 1427 cases and 1811 controls and have observed no significant association between *PD-1* rs2227981 polymorphism and the risk of cancer.

Nevertheless, the number of cases and controls as well as the number of polymorphisms in our meta-analysis is higher than in those previously published meta-analysis studies. 

It has been proposed that gene expression could be potentially affected by genetic polymorphisms [21,61,62,63]. Alterations in the expression of PD-1 and PDL-1 were detected in many cancer types including gastric cancer, lung cancer, thyroid cancer, laryngeal carcinoma, extrapulmonary small cell carcinoma, and breast cancer [63,64,65,66,67,68,69].

PD-1/PD-L1 axis impairs T cell activation by preventing Ras-Raf-MEK-ERK and PI3K-AKT signaling pathways, which are mainly believed to promote proliferation and differentiation of T cell [70]. The inhibitory regulation of PD-1/PD-L1 is typically compared to a brake in T cell activation [71]. PD-L1 is exerted by tumors to escape from immune system. Tumor-specific PD-L1-expression was not prognostic in colorectal cancer, while high immune cell-specific PD-1 expression was associated with a prolonged overall survival [72]. It has been revealed that high expression of PD-1 on peripheral blood T cell subsets is correlated with poor prognosis of metastatic gastric cancer [73]. Fang et al. [74] reported that the peripheral blood PD-1 expression was significantly higher in breast cancer patients than benign breast tumors. PD-1 and PD-L1 expression have been shown to be associated with adverse clinicopathological features in clear cell renal carcinoma [75].

This meta-analysis has however several limitations. Firstly, there are relatively small sample sizes of studies for some polymorphisms that should be expanded. Secondly, we have included in this meta-analysis only studies published in English, thus publication bias may have occurred. Thirdly, obvious heterogeneities were found in certain polymorphisms. Differences in ethnic background, type of cancer, and other baseline characteristics of participants may contribute to between-study heterogeneities. Lastly, gene-gene and gene-environment interactions which may affect cancer susceptibility were not evaluated in this meta-analysis due to lack of sufficient data. Therefore, the results of this meta-analysis should be cautiously interpreted.

In conclusion, the current meta-analysis suggests that rs2227981 and rs11568821 polymorphisms of *PD-1* and the rs4143815 polymorphism of *PD-L1* were associated with protection against cancer, while *PD-1* rs7421861 polymorphism significantly increased cancer risk. 

## 4. Methods

### 4.1. Literature Search

We searched PubMed, Web of Science, Scopus, and Google Scholar databases for publications that studied the association between *PD-1* and *PD-L1* polymorphisms and cancer risk. The last search was updated on 18 December 2019. The following search terms were used; “programmed cell death 1 or *PDCD1* or *PD-1*, or *CD279*, or programmed death-1-ligand 1 or *CD274* or *B7-H1*” and “polymorphism or single nucleotide polymorphism or SNP or variation” and “cancer or carcinoma, or tumor”. 

The process of recognizing eligible studies is presented in Figure 1. The inclusion and exclusion criteria were as follows. (1) The studies evaluated the association between the *PD-1* and *PD-L1* polymorphisms and cancer risk, (2) studies with necessary information on genotype or allele frequencies to estimate ORs and 95% Cis, (3) studies with human subjects, and (4) case-control design. We excluded reviews, conference papers, and other studies that were published as abstracts only.

### 4.2. Data Extraction

The data were recovered from eligible articles independently by two authors. Disagreements were discussed with the third investigator. The following information was recorded for each study: first author’s name, publication year, patient’s nationality, genotypes, and allele frequencies.

### 4.3. Statistical Analysis

We performed a meta-analysis to assess the association between *PD-1* and *PD-L1* polymorphisms and cancer susceptibility. The observed genotype frequencies in the controls were tested for Hardy-Weinberg equilibrium (HWE) using the chi-squared test. 

Odds ratio (OR) and 95% confidence interval (CI) were calculated to evaluate the association between *PD-1* and *PD-L1* polymorphisms and cancer risk in five genetic models, which were heterozygous, homozygous, dominant, recessive, and allele. The strength of the association between each polymorphism and cancer risk was assessed by pooled odds ratios (ORs) and their 95% confidence intervals (CIs). The Z-test was used for statistical significance of the pooled OR. We estimated the between-study heterogeneity by the Q-test and I2 test: If I2 < 50% and *P* > 0.1, the fixed effects model was used to estimate the ORs and the 95% CI; otherwise, the random effects model was applied. 

We evaluated publication bias using funnel plots for visual inspection and conducting quantitative estimations with Egger’s test. 

Sensitivity analysis was achieved by excluding each study in turn to assess the stability of the results. All analyses were achieved by STATA 14.1 software (Stata Corporation, College Station, TX, USA). 

## 5. Conclusions

The findings of our meta-analysis proposed that *PD-1* rs2227981, rs11568821, rs7421861, as well as *PD-L1* rs4143815 polymorphisms associated with overall cancer susceptibility. Further well-designed studies with large sample sizes are warranted to confirm our findings. 

## Figures and Tables

**Figure 1 cancers-11-01150-f001:**
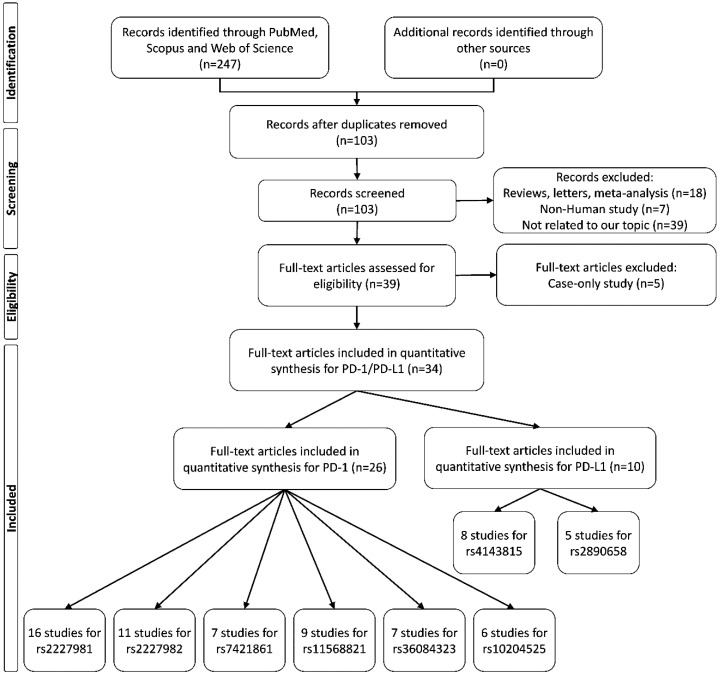
Flow diagram of study selection for this meta-analysis.

**Figure 2 cancers-11-01150-f002:**
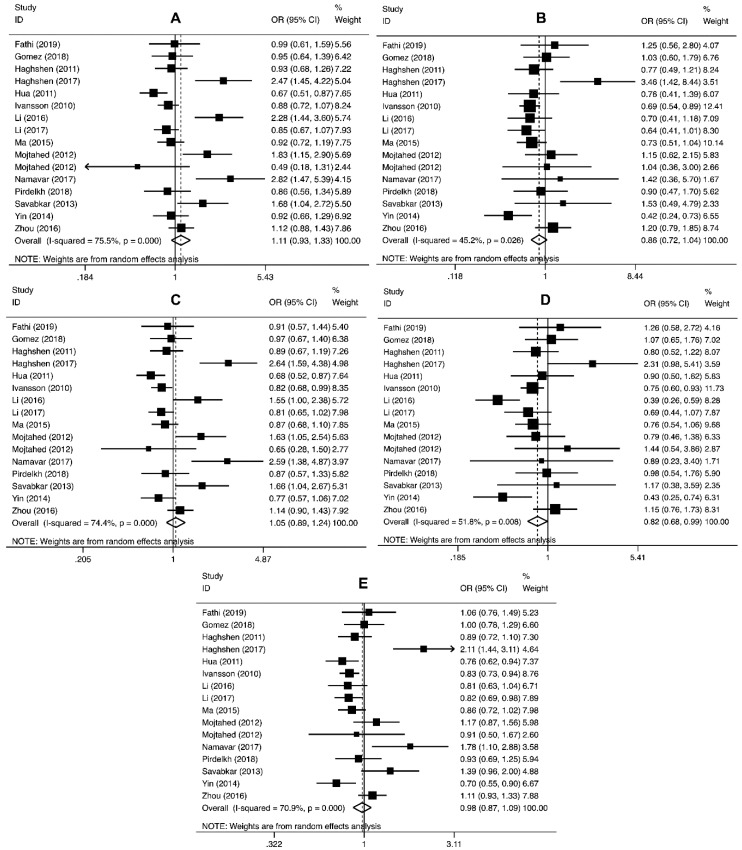
Forest plot for the association between *PD-1* rs2227981 polymorphism and cancer susceptibility for CT vs. CC (**A**), TT vs. CC (**B**), CT+TT vs. CC (**C**), TT vs. CT+TT (**D**), and T vs. C (**E**).

**Figure 3 cancers-11-01150-f003:**
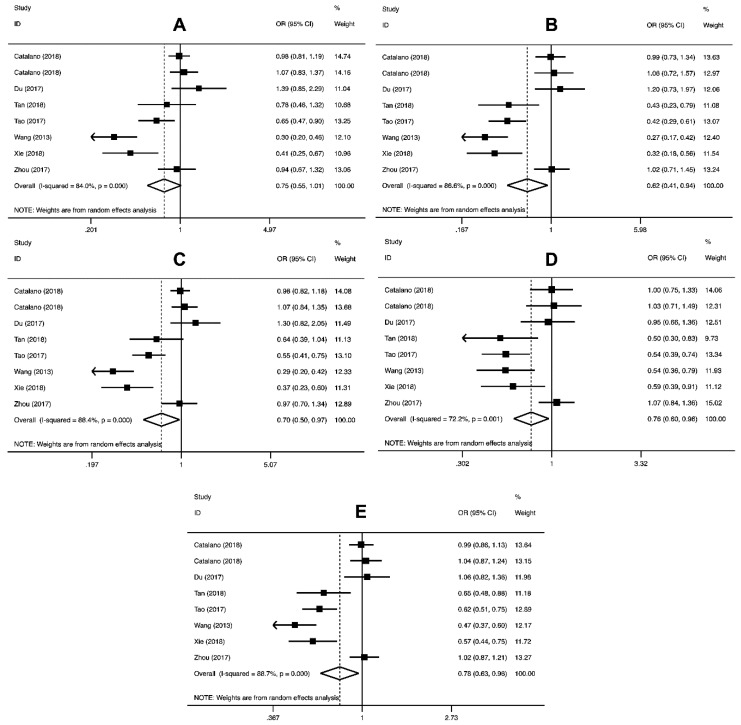
Forest plot of the relationship between *PD-L1* rs4143815 polymorphism and cancer susceptibility for CG vs. GG (**A**), CC vs. GG (**B**), CG+CC vs. GG (**C**), CC vs. CG+GG (**D**), and C vs. G (**E**).

**Figure 4 cancers-11-01150-f004:**
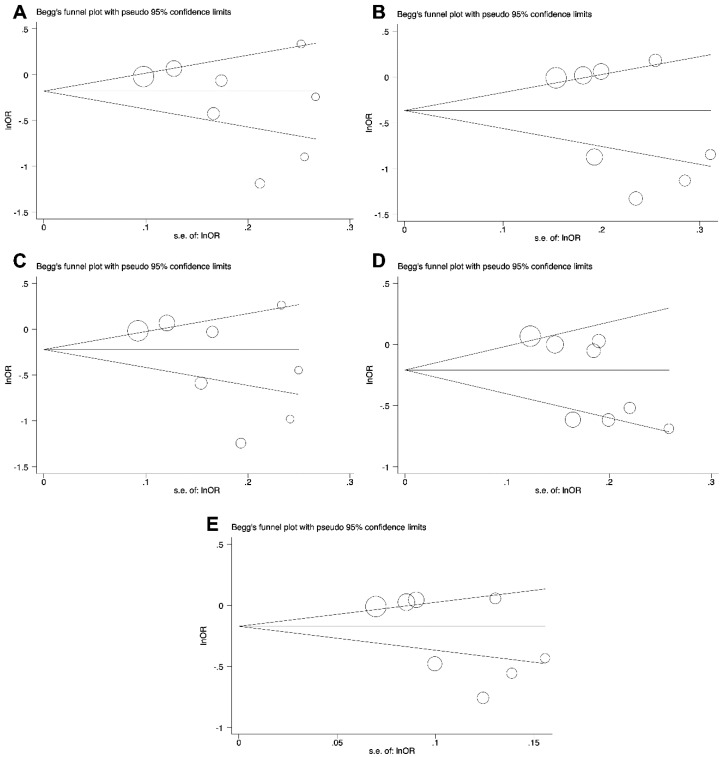
The funnel plot of *PD-L1* rs4143815 for the test of publication bias for CG vs. GG (**A**), CC vs. GG (**B**), CG+CC vs. GG (**C**), CC vs. CG+GG (**D**), and C vs. G (**E**).

**Figure 5 cancers-11-01150-f005:**
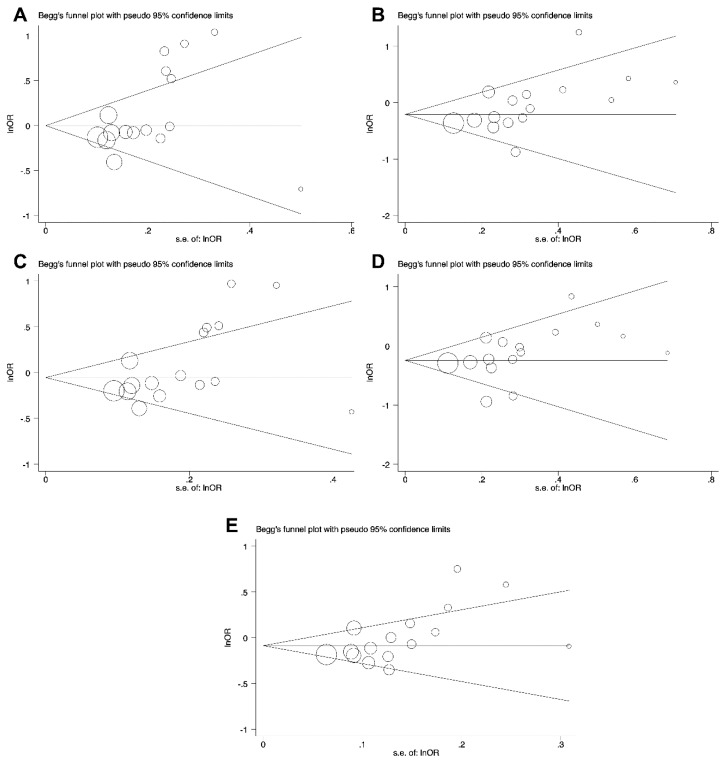
The funnel plot of *PD-1* rs2227981 polymorphism for the test of publication bias for CT vs. CC (**A**), TT vs. CC (**B**), CT+TT vs. CC (**C**), TT vs. CT+TT (**D**), and T vs. C (**E**).

**Figure 6 cancers-11-01150-f006:**
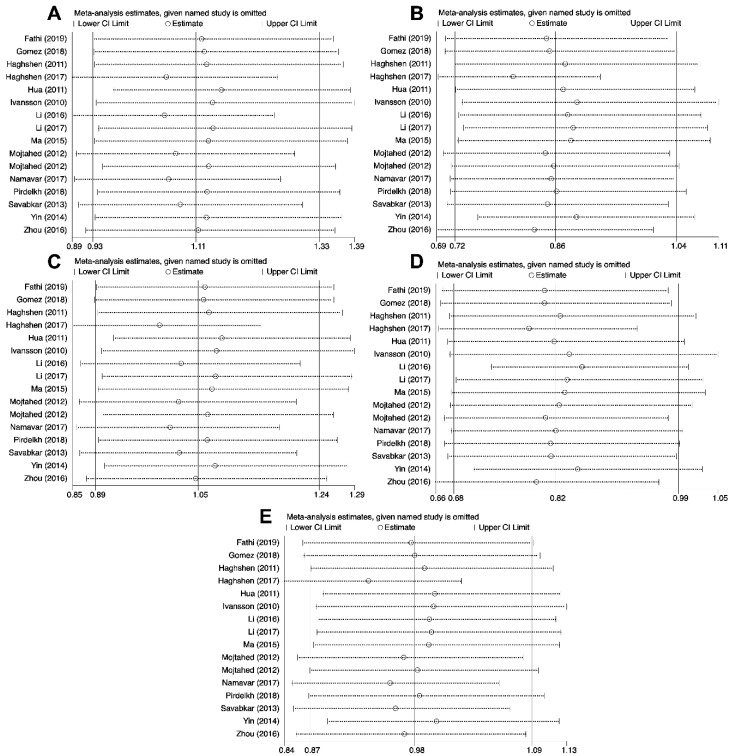
Sensitivity analyses for studies on *PD-1* rs2227981 polymorphism and cancer susceptibility for CG vs. GG (**A**), CC vs. GG (**B**), CG+CC vs. GG (**C**), CC vs. CG+GG (**D**), and C vs. G (**E**).

**Figure 7 cancers-11-01150-f007:**
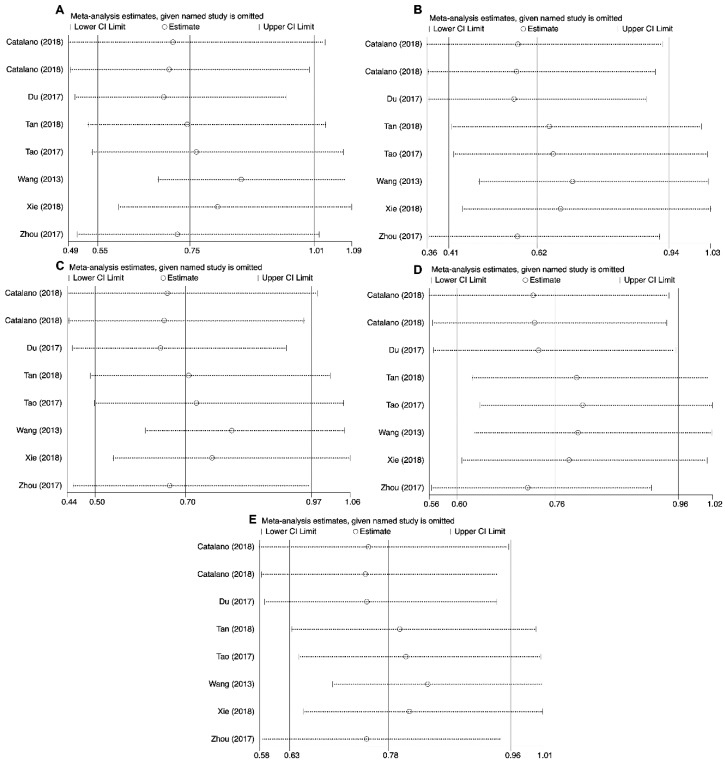
Sensitivity analyses for studies on *PD-L1* rs4143815 polymorphism and cancer susceptibility for CG vs. GG (**A**), CC vs. GG (**B**), CG+CC vs. GG (**C**), CC vs. CG+GG (**D**), and C vs. G (**E**).

**Table 1 cancers-11-01150-t001:** Locations and base pair positions of single nucleotide polymorphisms (SNPs) in *PD-1* and *PD-L1* genes.

Gene Name	db SNP rs # ID ^a^	Chromosome Position	Location	Base Change	Amino Acid Change
*PD-1*	rs2227981	chr2:241851121	Upstream	C/T	-
	rs2227982	chr2:241851281	Exon	C/T	Ala215Val
	rs7421861	chr2:241853198	Intron	T/C	-
	rs11568821	chr2:241851760	Intron	G/A	-
	rs36084323	chr2:241859444	Upstream	G/A	-
	rs10204525	chr2:241850169	3′UTR	A/G	-
*PD-L1*	rs4143815	chr9:5468257	3′UTR	G/C	-
	rs2890658	chr9:5465130	Intron	A/C	-

^a^ db = databases; rs # = reference SNP #; UTR: untranslated region.

**Table 2 cancers-11-01150-t002:** Characteristics of the studies eligible for meta-analysis.

First Author	Year	Country	Ethnicity	Cancer Type	Source of Control	Genotyping Method	Case/Control	Cases					Controls					HWE
*PD-1* rs2227981								CC	CT	TT	C	T	CC	CT	TT	C	T	
Fathi	2019	Iran	Asian	Squamous Cell Carcinomas of Head and Neck	HB	PCR-RFLP	150/150	65	69	16	199	101	66	71	13	203	97	0.317
Gomez	2018	Brazil	South American	Cutaneous Melanoma	PB	RT-PCR	250/250	87	126	37	300	200	85	130	35	300	200	0.188
Haghshenas	2011	Iran	Asian	Breast cancer	PB	PCR-RFLP	435/328	194	191	50	579	291	137	145	46	419	237	0.446
Haghshenas	2017	Iran	Asian	Thyroid cancer	PB	PCR-RFLP	105/160	40	51	14	131	79	99	51	10	249	71	0.331
Hua	2011	China	Asian	breast cancer	PB	PCR-RFLP	486/478	295	169	22	759	213	244	210	24	698	258	0.012
Ivansson	2010	Sweden	Caucasian	Cervical cancer	PB	TaqMan	1300/810	471	603	226	1545	1055	257	375	178	889	731	0.064
Li	2016	China	Asian	Cervical cancer	PB	PCR-RFLP	256/250	45	167	44	257	255	62	101	87	225	275	0.004
Li	2017	China	Asian	Ovarian cancer	HB	PCR-LDR	620/620	351	233	36	935	305	319	250	51	888	352	0.837
Ma	2015	China	Asian	lung cancer	PB	PCR-RFLP	528/600	244	216	68	704	352	256	246	98	758	442	0.004
Mojtahedi	2012	Iran	Asian	Colon cancer	PB	PCR-RFLP	175/200	47	102	26	196	154	75	89	36	239	161	0.290
Mojtahedi	2012	Iran	Asian	Rectal cancer	PB	PCR-RFLP	25/200	12	7	6	31	19	75	89	36	239	161	0.290
Namavar Jahromi	2017	Iran	Asian	Malignant Brain tumor	PB	PCR-RFLP	56/150	22	31	3	75	37	94	47	9	235	65	0.346
Pirdelkhosh	2018	Iran	Asian	NSCLC	PB	PCR-RFLP	206/173	78	100	28	256	156	60	89	24	209	137	0.321
Savabkar	2013	Iran	Asian	Gastric cancer	HB	PCR-RFLP	122/166	50	66	6	166	78	89	70	7	248	84	0.136
Yin	2014	China	Asian	Lung cancer	PB	PCR	324/330	198	106	20	502	146	181	105	44	467	193	0.001
Zhou	2016	China	Asian	ESCC	PB	PCR-LDR	584/585	291	241	52	823	345	310	229	46	849	321	0.683
*PD-1* rs2227982								CC	CT	TT	C	T	CC	CT	TT	C	T	
Fathi	2019	Iran	Asian	Squamous Cell Carcinomas of Head and Neck	HB	PCR-RFLP	150/150	146	4	0	296	4	146	4	0	296	4	0.868
Gomez	2018	Brazil	South American	Cutaneous Melanoma	PB	RT-PCR	250/250	227	21	2	475	25	225	25	0	475	25	0.405
Hua	2011	China	Asian	breast cancer	PB	PCR-RFLP	487/506	111	249	127	471	503	95	268	143	458	554	0.121
Ma	2015	China	Asian	lung cancer	PB	PCR-RFLP	528/600	343	148	37	834	222	404	168	28	976	224	0.056
Qiu	2014	China	Asian	esophageal cancer	HB	PCR-LDR	616/681	159	303	154	621	611	189	325	167	703	659	0.245
Ramzi	2018	Iran	Asian	Leukemia	PB	PCR-RFLP	59/38	38	18	3	94	24	17	19	2	53	23	0.255
Ren	2016	China	Asian	Breast Cancer	PB	MassARRAY	557/582	172	257	128	601	513	137	299	146	573	591	0.503
Tan	2018	China	Asian	Ovarian cancer	PB	PCR-RFLP	164/170	87	60	17	234	94	111	48	11	270	70	0.075
Tang	2015	China	Asian	Gastric cardia adenocarcinoma	HB	PCR-LDR	330/603	75	168	87	318	342	163	292	148	618	588	0.448
Tang	2017	China	Asian	Esophagogastric junction adenocarcinoma	HB	SNPscan	1041/1674	220	549	272	989	1093	416	816	442	1648	1700	0.309
Zhou	2016	China	Asian	ESCC	PB	PCR-LDR	584/585	149	305	130	603	565	150	297	138	597	573	0.702
*PD-1* rs7421861								TT	TC	CC	T	C	TT	TC	CC	T	C	
Ge	2015	China	Asian	Colon cancer	HB	PCR-RFLP	199/620	133	60	6	326	72	440	163	17	1043	197	0.685
Ge	2015	China	Asian	Rectal cancer	HB	PCR-RFLP	362/620	241	114	7	596	128	440	163	17	1043	197	0.685
Hua	2011	China	Asian	Breast cancer	PB	PCR-RFLP	490/512	333	146	11	812	168	370	130	12	870	154	0.885
Qiu	2014	China	Asian	esophageal cancer	HB	PCR-LDR	600/673	411	168	21	990	210	460	188	25	1108	238	0.295
Ren	2016	China	Asian	Breast Cancer	PB	MassARRAY	560/580	341	196	23	878	242	347	205	28	899	261	0.746
Tang	2015	China	Asian	Gastric cardia adenocarcinoma	HB	PCR-LDR	324/598	226	91	7	543	105	408	168	22	984	212	0.368
Tang	2017	China	Asian	esophagogastric junction adenocarcinoma	HB	SNPscan	1041/1674	642	358	41	1642	440	1166	454	54	2786	562	0.232
*PD-1* rs11568821								GG	AG	AA	G	A	GG	AG	AA	G	A	
Bayram	2012	Turkey	Asian	liver cancer	HB	PCR-RFLP	236/236	191	45	0	427	45	180	56	0	416	56	0.039
Fathi	2019	Iran	Asian	Squamous Cell Carcinomas of Head and Neck	HB	PCR-RFLP	150/150	119	27	4	265	35	113	32	5	258	42	0.162
Haghshenas	2011	Iran	Asian	Breast cancer	PB	PCR-RFLP	436/290	365	63	8	793	79	231	55	4	517	63	0.726
Haghshenas	2017	Iran	Asian	Thyroid cancer	PB	PCR-RFLP	95/160	82	13	0	177	13	127	30	3	284	36	0.440
Ma	2015	China	Asian	lung cancer	PB	PCR-RFLP	528/600	426	102	0	954	102	456	142	2	1054	146	0.009
Namavar Jahromi	2017	Iran	Asian	Malignant Brain tumor	PB	PCR-RFLP	56/150	47	8	1	102	10	116	30	4	262	38	0.240
Pirdelkhosh	2018	Iran	Asian	NSCLC	PB	PCR-RFLP	206/173	171	31	4	373	39	144	26	3	314	32	0.168
Ramzi	2018	Iran	Asian	Leukemia	PB	PCR-RFLP	59/38	38	18	3	94	24	21	13	4	55	21	0.373
Yousefi	2013		Asian	colon cancer	PB		80/110	18	27	35	63	97	43	45	22	131	89	0.114
*PD-1* rs36084323								GG	AG	AA	G	A	GG	AG	AA	G	A	
Gomez	2018	Brazil	South American	Cutaneous Melanoma	PB	RT-PCR	250/250	226	18	6	470	30	225	25	0	475	25	0.405
Hua	2011	China	Asian	Breast cancer	PB	PCR-RFLP	490/512	103	271	116	477	503	140	260	112	540	484	0.673
Li	2017	China	Asian	Ovarian cancer	HB	PCR-LDR	620/620	150	301	169	601	639	168	323	129	659	581	0.251
Ma	2015	China	Asian	lung cancer	PB	PCR-RFLP	528/600	144	246	138	534	522	156	296	148	608	592	0.747
Shamsdin	2018	Iran	Asian	Colon cancer	PB	PCR-RFLP	76/73	60	15	1	135	17	18	28	27	64	82	0.059
Tang	2017	China	Asian	esophagogastric junction adenocarcinoma	HB	SNPscan	1041/1674	238	521	282	997	1085	430	800	444	1660	1688	0.071
Zhou	2016	China	Asian	ESCC	PB	PCR-LDR	584/585	147	303	134	597	571	145	298	142	588	582	0.649
*PD-1* rs10204525								AA	AG	GG	A	G	AA	AG	GG	A	G	
Li	2013	China	Asian	HCC	PB	TIANamp	271/318	180	83	8	443	99	160	130	28	450	186	0.828
Qiu	2014	China	Asian	esophageal cancer	HB	PCR-LDR	600/651	317	240	43	874	326	345	243	63	933	369	0.039
Ren	2016	China	Asian	Breast Cancer	PB	MassARRAY	559/582	257	248	54	762	356	291	240	51	822	342	0.880
Tang	2015	China	Asian	Gastric cardia adenocarcinoma	HB	PCR-LDR	313/581	169	123	21	461	165	309	219	53	837	325	0.120
Tang	2017	China	Asian	esophagogastric junction adenocarcinoma	HB	SNPscan	1039/1674	544	397	98	1485	593	870	672	132	2412	936	0.888
Zhou	2016	China	Asian	ESCC	PB	PCR-LDR	584/585	325	226	33	876	292	296	238	51	830	340	0.749
*PD-L1* rs4143815								GG	CG	CC	G	C	GG	CG	CC	G	C	
Catalano	2018	Czech	Caucasian	Colon cancer	HB	TaqMan	824/1103	388	345	91	1121	527	514	467	122	1495	711	0.306
Catalano	2018	Czech	Caucasian	Rectal cancer	HB	TaqMan	371/1103	167	162	42	496	246	514	467	122	1495	711	0.306
Du	2017	China	Asian	NSCLC	HB	sequencing	320/199	52	145	123	249	391	40	80	79	160	238	0.021
Tan	2018	China	Asian	Ovarian cancer	PB	PCR-RFLP	164/170	51	82	31	184	144	38	78	54	154	186	0.334
Tao	2017	China	Asian	Gastric cancer	HB	Sequencing	346/500	123	153	70	399	293	117	223	160	457	543	0.023
Wang	2013	China	Asian	Gastric cancer	HB	sequencing	205/393	88	72	45	248	162	70	188	135	328	458	0.746
Xie	2018	China	Asian	HCC	HB	sequencing	225/200	74	101	50	249	201	31	104	65	166	234	0.316
Zhou	2017	China	Asian	ESCC	PB	PCR-LDR	575/577	87	277	211	451	699	85	289	203	459	695	0.275
PD-L1 rs2890658								AA	AC	CC	A	C	AA	AC	CC	A	C	
Chen	2014	China	Asian	NSCLC	HB	PCR-RFLP	293/293	242	48	3	532	54	266	26	1	558	28	0.671
Cheng	2015	China	Asian	NSCLC	HB	PCR-RFLP	288/300	233	51	4	517	59	269	30	1	568	32	0.867
Ma	2015	China	Asian	lung cancer	PB	PCR-RFLP	528/600	416	106	6	938	118	512	84	4	1108	92	0.785
Xie	2018	China	Asian	HCC	HB	sequencing	225/200	170	49	6	389	61	139	55	6	333	67	0.844
Zhou	2017	China	Asian	ESCC	PB	PCR-LDR	575/577	18	161	396	197	953	15	144	418	174	980	0.541

List of Abbreviations: HCC: Hepatocellular carcinoma; PB: Population-based; HB: Hospital-based; ESCC: Esophageal squamous cell carcinoma; LDR: Ligase Detection Reaction; NSCLC: non-small cell lung cancer; PCR-RFLP: PCR-Restriction fragment length polymorphism; HWE: Hardy-Weinberg equilibrium; MassARRAY^®^System: Nonfluorescent detection platform utilizing mass spectrometry to accurately measure PCR-derived amplicons.

**Table 3 cancers-11-01150-t003:** The pooled ORs and 95% CIs for the association between *PD-1* and *PD-L1* polymorphisms and cancer susceptibility.

Polymorphism	*n*	Genetic Model	Association Test			Heterogeneity Test			Publication Bias Test	
			OR (95% CI)	Z	*p*	χ^2^	I^2^ (%)	*p*	Egger’s Test *p*	Begg’s Test *p*
*PD-1* rs2227981	16	CT vs. CC	1.11 (0.93–1.33)	1.16	0.25	61.22	75	<0.00001	0.032	0.031
		TT vs. CC	0.86 (0.72–1.04)	1.51	0.13	27.39	45	0.03	0.034	0.024
		CT+TT vs. CC	1.05 (0.89–1.24)	0.64	0.52	58.58	74	<0.00001	0.019	0.005
		TT vs. CT+CC	0.82 (0.68–0.99)	2.04	0.04	31.12	52	0.008	0.155	0.150
		T vs. C	0.98 (0.87–1.09)	0.43	0.66	51.48	71	<0.00001	0.020	0.012
*PD-1* rs2227982	11	CT vs. CC	1.01 (0.85–1.19)	0.09	0.930	24.53	59	0.006	0.359	0.186
		TT vs. CC	1.05 (0.87–1.26)	0.51	0.611	17.10	47	0.050	0.288	0.180
		CT+TT vs. CC	1.02 (0.86–1.20)	0.22	0.829	26.49	62	0.003	0.469	0.484
		TT vs. CT+CC	1.00 (0.90–1.10)	0.04	0.97	7.52	0	0.581	0.184	0.211
		T vs. C	1.02 (0.92–1.12)	0.38	0.707	20.50	51	0.025	0.927	0.715
*PD-1* rs11568821	9	AG vs. GG	0.79 (0.67–0.94)	2.73	0.006	3.89	0	0.87	0.499	0.409
		AA vs. GG	1.01 (0.47–2.14)	0.01	0.99	13.19	47	0.07	0.015	0.091
		AG+AA vs. GG	0.82 (0.70–0.96)	2.42	0.020	11.30	29	0.19	0.613	0.835
		AA vs. AG+GG	1.07 (0.54–2.13)	0.19	0.846	11.79	41	0.11	0.010	0.095
		A vs. G	0.88 (0.68–1.15)	0.92	0.36	24.39	67	0.002	0.822	0.835
*PD-1* rs7421861	7	CT vs. TT	1.16 (1.02–1.33)	2.20	0.03	0.01	46	0.09	0.215	0.881
		CC vs. TT	1.00 (0.79–1.28)	0.03	0.98	4.76	0	0.57	0.116	0.881
		CT+CC vs. TT	1.14 (0.99–1.31)	1.81	0.07	12.93	54	0.04	0.196	0.453
		CC vs. CT+TT	0.96 (0.75–1.22)	0.37	0.71	3.49	0	0.75	0.101	0.652
		C vs. T	1.09 (0.97–1.23)	1.42	0.16	13.02	54	0.04	0.200	0.652
*PD-1* rs36084323	7	AG vs. GG	0.92 (0.71–1.20)	0.60	0.55	27.83	78	0.0001	0.042	0.051
		AA vs. GG	1.08 (0.77–1.52)	0.45	0.66	28.21	79	0.0001	0.079	0.188
		AG+AA vs. GG	0.88 (0.64–1.21)	0.79	0.43	47.46	87	<0.00001	0.081	0.293
		AA vs. AG+GG	1.06 (0.83–1.36)	0.46	0.64	22.86	74	0.0008	0.137	0.348
		A vs. G	0.89 (0.70–1.14)	0.92	0.36	66.01	91	<0.00001	0.160	0.453
*PD-1* rs10204525	6	AG vs. AA	0.94 (0.80–1.10)	0.76	0.45	13.13	62	0.02	0.640	0.851
		GG vs. AA	0.76 (0.53–1.09)	1.48	0.14	19.40	74	0.002	0.031	0.091
		AG+GG vs. AA	0.90 (0.75–1.08)	1.10	0.27	18.41	73	0.002	0.399	0.188
		GG vs. AG+AA	0.78 (0.57–1.09)	1.46	0.14	16.64	70	0.005	0.020	0.039
		G vs. A	0.89 (0.76–1.05)	1.38	0.17	23.71	79	0.0002	0.172	0.091
*PD-L1* rs4143815	8	CG vs. GG	0.75 (0.55–1.01)	1.89	0.06	43.76	84	<0.0001	0.230	0.322
		CC vs. GG	0.62 (0.41–0.94)	2.28	0.02	52.19	87	<0.00001	0.188	0.138
		CG+CC vs. GG	0.70 (0.50–0.97)	2.15	0.03	43.20	84	<0.00001	0.184	0.138
		CC vs. CG+GG	0.76 (0.60–0.96)	2.30	0.02	25.19	72	0.0007	0.070	0.138
		C vs. G	0.78 (0.63–0.96)	2.33	0.02	61.68	89	<0.00001	0.100	0.138
*PD-L1* rs2890658	5	AC vs. AA	1.36 (0.92–2.01)	1.53	0.13	13.83	71	0.008	0.757	0.624
		CC vs. AA	1.12 (0.68–1.84)	0.45	0.65	4.31	7	0.37	0.032	0.050
		AC+CC vs. AA	1.35 (0.89–2.04)	1.43	0.15	16.24	75	0.003	0.736	1.000
		CC vs. AC+AA	0.90 (0.71–1.15)	0.83	0.41	4.25	6	0.37	0.041	0.050
		C vs. A	1.30 (0.88–1.91)	1.32	0.19	25.96	85	<0.0001	0.248	0.142

**Table 4 cancers-11-01150-t004:** Stratified analysis of *PD-1* and *PD-L1* polymorphisms with cancer susceptibility.

Variable	No.	CT vs. CC		TT vs. CC		CT+TT vs. CC		TT vs. CT+CC		T vs. C	
***PD-1*** rs2227981		OR (95% CI)	*p*	OR (95% CI)	*p*	OR (95% CI)	*p*	OR (95% CI)	*p*	OR (95% CI)	*p*
Asian	14	1.16 (0.94–1.43)	0.173	0.89 (0.71–1.12)	0.312	1.09 (0.90–1.32)	0.393	0.83 (0.66–1.04)	0.106	1.00 (0.87–1.14)	0.953
Population-based	13	1.12 (0.91–1.39)	0.276	0.88 (0.70–1.07)	0.175	1.06 (0.87–1.28)	0.571	0.81 (0.66–1.01)	0.060	0.97 (0.85–1.10)	0.611
Hospital-based	3	1.06 (0.72–1.61)	0.714	0.91 (0.53–1.59)	0.749	1.04 (0.68–1.57)	0.873	0.85 (0.57–1.26)	0.421	1.03 (0.76–1.41)	0.839
Gastrointestinal cancer	3	1.13 (0.73–1.76)	0.588	0.68 (0.56–0.84)	0.000	0.95 (0.71–1.27)	0.713	0.60 (0.40–0.89)	0.011	0.83 (0.75–0.91)	0.000
Lung cancer	3	0.91 (0.76–1.10)	0.324	0.65 (0.44–0.97)	0.030	0.84 (0.71–0.99)	0.043	0.69 (0.45–1.04)	0.079	0.83 (0.72–0.95)	0.009
Breast cancer	2	0.78 (0.56–1.08)	0.136	0.76 (0.53–1.10)	0.147	0.80 (0.59–1.01)	0.058	0.83 (0.59–1.17)	0.291	0.82 (0.70–0.96)	0.012
***PD-1*** rs2227982		CT vs. CC		TT vs. CC		CT+TT vs. CC		TT vs. CT+CC		T vs. C	
Asian	10	1.02 (0.85–1.21)	0.845	1.04 (0.87–1.26)	0.655	1.02 (0.86–1.22)	0.790	1.00 (0.90–1.10)	0.921	1.02 (0.92–1.12)	0.708
Population-based	8	0.91 (0.73–1.12)	0.363	0.99 (0.73–1.33)	0.934	0.93 (0.741.16)	0.507	0.99 (0.83–1.17)	0.734	0.98 (0.85–1.14)	0.818
Hospital-based	3	1.22 (1.06–1.40)	0.006	1.16 (0.99–1.37)	0.067	1.20 (1.05–1.37)	0.008	1.02 (0.89–1.16)	0.806	1.08 (0.99–1.17)	0.077
Gastrointestinal cancer	4	1.18 (1.04–1.34)	0.011	1.12 (0.97–1.29)	0.133	1.16 (1.03–1.30)	0.017	1.00 (0.89–1.12)	0.989	1.06 (0.98–1.34)	0.146
Breast cancer	2	0.73 (0.59–0.90)	0.004	0.73 (0.57–0.93)	0.010	0.73 (0.60–0.89)	0.002	0.89 (0.74–1.09)	0.257	0.85 (0.76–0.96)	0.010
***PD-1*** rs7421861		CT vs. TT		CC vs. TT		CT+CC vs. TT		CC vs. CT+TT		C vs. T	
Hospital-based	5	1.89 (1.01–1.40)	0.042	1.05 (0.79–1.39)	0.745	1.16 (0.98–1.38)	0.096	0.99 (0.74–1.31)	0.916	1.11 (0.95–1.29)	0.192
Population-based	2	1.09 (0.86–1.39)	0.478	0.89 (0.56–1.43)	0.630	1.07 (0.84–1.37)	0.565	0.88 (0.55–1.40)	0.586	1.04 (0.85–1.28)	0.692
Gastrointestinal cancer	5	1.19 (1.01–1.40)	0.042	1.05 (0.79–1.39)	0.745	1.16 (0.97–1.38)	0.096	1.00 (0.75–1.32)	0.979	1.11 (0.95–1.29)	0.192
Breast cancer	2	1.09 (0.86–1.39)	0.478	0.89 (0.56–1.43)	0.630	1.07 (0.84–1.37)	0.565	0.88 (0.55–1.40)	0.586	1.04 (0.85 (1.28)	0.692
***PD-1*** rs11568821		AG vs. GG		AA vs. GG		AG+AA vs. GG		AA vs. AG+GG		A vs. G	
Population-based	7	0.80 (0.66–0.97)	0.020	1.02 (0.43–2.42)	0.968	0.86 (0.65–1.14)	0.288	1.09 (0.50–2.38)	0.833	0.92 (0.78–1.08)	0.294
Hospital-based	2	0.77 (0.55–1.10)	0.150	0.76 (0.20–2.90)	0.688	0.77 (0.55–1.09)	0.140	0.76 (0.21–3.02)	0.736	0.80 (0.58–1.09)	0.152
***PD-1*** rs36084323		AG vs. GG		AA vs. GG		AG+AA vs. GG		AA vs. AG+GG		A vs. G	
Asian	6	0.95 (0.71–1.25)	0.691	1.05 (0.75–1.47)	0.769	0.86 (0.61–1.23)	0.412	1.05 (0.82–1.33)	0.715	0.86 (0.67–1.12)	0.259
Population-based	5	0.78 (0.50–1.21)	0.268	0.88 (0.47–1.63)	0.674	0.71 (0.42–1.22)	0.219	0.94 (0.61–1.45)	0.767	0.74 (0.49–1.12)	0.152
Hospital–based	2	1.13 (0.97–1.32)	0.127	1.26 (1.00–1.59)	0.052	1.17 (1.01–1.35)	0.042	1.93 (0.87–1.64)	0.277	1.12 (1.00–1.26)	0.05
***PD-1*** rs10204525		AG vs. AA		GG vs. AA		AG+GG vs. AA		GG vs. AG+AA		G vs. A	
Gastrointestinal cancer	5	0.90 (0.76–1.07)	0.227	0.63 (0.45–1.04)	0.078	0.86 (0.70–1.04)	0.121	0.72 (0.48–1.06)	0.096	0.85 (0.71–1.02)	0.077
Population-based	3	0.85 (0.58–1.23)	0.382	0.60 (0.28–1.32)	0.203	0.80 (0.52–1.32)	0.312	0.65 (0.35–1.23)	0.186	0.80 (0.55–1.17)	0.246
Hospital-based	3	0.99 (0.88–1.22)	0.908	0.90 (0.63–1.29)	0.568	0.99 (0.88–1.11)	0.831	0.89 (0.60–1.32)	0.560	0.99 (0.90–1.08)	0.767
***PD-L1*** rs4143815		CG vs. GG		CC vs. GG		CG+CC vs. GG		CC vs. CG+GG		C vs. G	
Gastrointestinal cancer	6	0.68 (0.48–0.97)	0.032	0.59 (0.37–0.96)	0.033	0.64 (0.43–0.95)	0.028	0.77 (0.58–1.02)	0.064	0.76 (0.59–0.98)	0.034
Hospital-based	6	0.71 (0.48–1.05)	0.087	0.60 (0.36–1.00)	0.051	0.67 (0.44–1.02)	0.059	0.75 (0.58–0.97)	0.030	0.76 (0.58–0.99)	0.043
Population-based	2	0.89 (0.68–1.18)	0.414	0.68 (0.29–1.59)	0.378	0.82 (0.55–1.23)	0.332	0.76 (0.36–1.59)	0.460	0.83 (0.53–1.30)	0.413
***PD-L1*** rs2890658		AC vs. AA		CC vs. AA		AC+CC vs. AA		CC vs. AC+AA		C vs. A	
Lung cancer	3	1.74 (1.37–2.19)	0.000	2.48 (0.92–6.69)	0.072	1.77 (1.41–2.23)	0.000	2.29 (0.85–6.16)	0.101	1.72 (1.39–2.13)	0.000
Gastrointestinal cancer	2	4.34 (0.13–148.07)	0.415	4.43 (0.17–112.70)	0.368	0.76 (0.53–1.10)	0.141	0.84 (0.66–1.08)	0.179	0.84 (0.69–1.01)	0.070
Hospital-based	3	1.42 (0.72–2.96)	0.317	1.61 (0.52–4.98)	0.409	1.45 (0.72–2.92)	0.296	1.45 (0.57–3.73)	0.439	1.46 (0.75–2.82)	0.266
Population-based	2	6.30 (0.39–103.18)	0.197	6.85 (0.60–78.36)	0.122	1.23 (0.67–2.26)	0.503	0.90 (0.56–1.37)	0.636	1.13 (0.65–1.97)	0.661

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
