# Peer review of "Association between PD-1 and PD-L1 Polymorphisms and the Risk of Cancer: A Meta-Analysis of Case-Control Studies"

_cancers, 2019, doi:10.3390/cancers11081150_

Round 1
Reviewer 1 Report
In this study, Hasehmi and collaborators perform a comprehensive meta-analysis to assess the association of PD-1 and PD-L1 polymorphisms with the risk of cancer. Although the manuscript is well written and study is well conducted some critical points need to be clarified.
Introduction needs to be improved. Authors have to develop and detail the chromosomal localization of PD-1, PD-L1, the localization of the polymorphisms studied, the functional consequence of these polymorphisms in term of protein function etc
Two key publications have to be cited, introduced and discussed as they also explore the association of PD-1 polymorphism with the risk of cancer and present contradictory results. Dong et al, Plos One 2016; Mamat et al, Int J Clin Exp Med 2015.
Finally, what’s about the association of PD-1 and PD-L1 polymorphisms with the risk of cancer type. Indeed, Mamat et al evokes an association of rs2227981 with digestive tumor and tumor occurring in woman.
Other points
- Figure 2, 3, 5, 6: letters (A to E) are missing
- Figure 2-7 are blurred
- Abbreviation used in Table 1 are needed
Author Response
In this study, Hashemi and collaborators perform a comprehensive meta-analysis to assess the association of PD-1 and PD-L1 polymorphisms with the risk of cancer. Although the manuscript is well written and study is well conducted some critical points need to be clarified.
Introduction needs to be improved. Authors have to develop and detail the chromosomal localization of PD-1, PD-L1, the localization of the polymorphisms studied, the functional consequence of these polymorphisms in term of protein function etc
Author response: We appreciate respected reviewer comment. We have added a table (Table 1) to the introduction which identify the chromosome position, location and base changes as well as amino acid changes of polymorphisms in PD-1 and PD-L1 (Page 2 of revised manuscript).
Two key publications have to be cited, introduced and discussed as they also explore the association of PD-1 polymorphism with the risk of cancer and present contradictory results. Dong et al, Plos One 2016; Mamat et al, Int J Clin Exp Med 2015.
Author response: We appreciate the respected reviewer comment. We have added reference 59 (Dong et al) and 60 (Mamat et al) in the revised version of manuscript and discussed it in the revised “Discussion” section (p16, Line 244-252).
Finally, what’s about the association of PD-1 and PD-L1 polymorphisms with the risk of cancer type. Indeed, Mamat et al evokes an association of rs2227981 with digestive tumor and tumor occurring in woman.
Author response: To address the respected reviewer thoughtful comments we performed stratified analysis and the results are presented in Table 4 (page 10 and 11) and result section (page 10, Line: 118-137).
Other points
- Figure 2, 3, 5, 6: letters (A to E) are missing
- Figure 2-7 are blurred
- Abbreviation used in Table 1 are needed
Author response: Thank you for careful comments. We corrected and addressed all issues. We also include original high quality figures as supplementary so they are available for readers and reviewers.
Reviewer 2 Report
The authors identified polymorphisms in PD-1 and PD-L1 that can predict cancer risks based on an meta-analysis study.
Introduction
*In the introduction, the rationale/relevance/added value of the study is missing. This has to be better explained.
*Furthermore, is it possible to explain in more detail the correlation between SNPs and cancer risks? Some examples. This would also render more rationale for the added value of this study.
Results + figures:
* In the flow diagram (figure 1) please make a clear difference between PD-1 and PD-L1 polymorphisms. Indicate in the diagram which are PD-1 and PD-L1 polymorphisms. As such, the reader is able to understand the figure without reading the text. In this context, it would be as well recommended to expand the legends of each figure. The figures need to be clear by reading the figure legend.
* Figures 2-7 are not from good quality, this had te be improved.
Discussion:
*Please highlight more the added value and relevance of the study in the discussion.
*Furthermore, please provide some insight in the "function/role" of the PD-1/PD-L1 polymorphisms that are related with cancer risks. Provide some hypothesize why these polymorphisms are related with cancer risk.
*Line 195: is there a link with the Ras-Raf-Mek-ERK and PI3K-AKT pathway and this polymorphisms? Not clear why it is cited here.
*Discuss more in detail the role of different biomarkers and immune checkpoint therapy outcome.
*Added value of identifying these polymorphisms is missing.
*Line 208: the authors claim themselves that his analysis has several limitations, including that they found heterogeneities in certain polymorphisms because they didn't look in detail to the type of cancer. Since PD-1/PD-L1 therapy outcome shows a high variable outcome between different cancer types, this is an important factor and I think it is necessary to include this factor in the study.
*The authors have identified new biomarkers. But is it possible to propose some therapies for cancer patients with these polymorphisms in the discussion?
Author Response
Comments and Suggestions for Authors
The authors identified polymorphisms in PD-1 and PD-L1 that can predict cancer risks based on an meta-analysis study.
Introduction
*In the introduction, the rationale/relevance/added value of the study is missing. This has to be better explained.
Author response: We appreciate the respected review feedback. Actually we the importance of study was described in original text of our manuscript. We have highlighted with red font in revised manuscript. We have carefully explained the important of PD-1 and PD-L1 in cancer and cell death (Page 2, Line 49-67).
*Furthermore, is it possible to explain in more detail the correlation between SNPs and cancer risks? Some examples. This would also render more rationale for the added value of this study.
Author response: We appreciate the respected review feedback. And added some of the major cancers that the correlation with SNP of PD-1 and PD-L1 has been investigation (page 2, line 67-68).
Results + figures:
* In the flow diagram (figure 1) please make a clear difference between PD-1 and PD-L1 polymorphisms. Indicate in the diagram which are PD-1 and PD-L1 polymorphisms. As such, the reader is able to understand the figure without reading the text. In this context, it would be as well recommended to expand the legends of each figure. The figures need to be clear by reading the figure legend.
* Figures 2-7 are not from good quality, this had to be improved.
Author response: Thanks for feedback. We revised the flow diagram and improved the quality of figures and also included all original high quality figures as supplementary so they are available for the readers and reviewers.
Discussion:
*Please highlight more the added value and relevance of the study in the discussion.
Author response: We appreciate the respected reviewer feedback. We have highlighted the importance and relevance (Line 220-225) after rephrasing our statement I the discussion and highlighting the importance of our current investigation.
*Furthermore, please provide some insight in the "function/role" of the PD-1/PD-L1 polymorphisms that are related with cancer risks. Provide some hypothesize why these polymorphisms are related with cancer risk.
Author response: We appreciate the respected reviewer comment. We have highlighted the importance and relevance (Line 220-225) after rephrasing/editing of our statement in the discussion and highlighting the importance of our current investigation.
*Line 195: is there a link with the Ras-Raf-Mek-ERK and PI3K-AKT pathway and this polymorphisms? Not clear why it is cited here.
Author Response: Thanks for reviewer thoughtful comment. Here we explained the mechansims of action of PD-1/PD-L1 axis in different cells. As we have mentioned PD-1/PD-L1 impairs T cell activation via the Ras-Raf-Mek-ERK and PI3K-AKT pathway. The link between polymorphisms and this pathways should be investigated in a study which consider activity of PD-1/PD-L1 and different polymorphism (or functional polymorphisms).
*Discuss more in detail the role of different biomarkers and immune checkpoint therapy outcome.
Author Response: The focus of present study was not the biomarkers and checkpoint therapy. Although we appreciate the respected reviewer comments we leave it for our future studies.
*Added value of identifying these polymorphisms is missing.
Author response: We appreciate the respected reviewer comment. Actually the value of identifying the polymorphism has been widely discussed in the introduction (line 49-66). Over there we have discussed in details about the importance of PD-1 and PD-L1 in cell death and then in cancer and in the end we have mentioned the importance of the current study.
*Line 208: the authors claim themselves that his analysis has several limitations, including that they found heterogeneities in certain polymorphisms because they didn't look in detail to the type of cancer. Since PD-1/PD-L1 therapy outcome shows a high variable outcome between different cancer types, this is an important factor and I think it is necessary to include this factor in the study.
Author response: Thanks for your comments. Subgroup analyses was done and the results is presented in Table 4 and result section.
*The authors have identified new biomarkers. But is it possible to propose some therapies for cancer patients with these polymorphisms in the discussion?
Author response: We appreciate the respected reviewer comment. The current study was not focused on polymorphisms which affect activity in apoptosis therefore we are not able to comment about therapeutic application of the current study.
Reviewer 3 Report
I have read with interest the study "Association between PD-1 and PD-L1 polymorphisms and the risk of cancer: A meta-analysis of case-control studies" from Hashemi et al. and I find it of potential interest for the readers of Cancers.
I was positively impressed from the thoroughness of the analyses and I only have minor comments that I think should be addressed before supporting the publication of this article.
- line 28: typo PDL-1 should be PD-L1
- The figures have quite low resolution, I think should be provided in higher quality. sometimes it is even difficult to read the names of the authors of the studies in, for example, fig 1 and 3. Also the panels miss the letters
- Despite I found the study very sound, I think the discussion has quite some room for improvement. my two suggestions are 1) discuss further about potential mechanisms by which polymorphism in PD-1/PD-L1 can affect cancer susceptibility. e.g. do those changes result in change in the amino acid sequence of the protein? Is there any study that have correlated SNP with gene expression? are those codons predicted to have different translation efficiency? 2) I found intriguing that some of the mentioned studies have very distinct results from the others. I think this observation deserves a comment and an attempt of explaining the inconsistency (e.g. Haghshenas et al 2017 is a clear outlier in fig 1)
Author Response
I have read with interest the study "Association between PD-1 and PD-L1 polymorphisms and the risk of cancer: A meta-analysis of case-control studies" from Hashemi et al. and I find it of potential interest for the readers of Cancers.
I was positively impressed from the thoroughness of the analyses and I only have minor comments that I think should be addressed before supporting the publication of this article.
- line 28: typo PDL-1 should be PD-L1
Author response: We appreciate the respected reviewer comment. We corrected it.
- The figures have quite low resolution, I think should be provided in higher quality. sometimes it is even difficult to read the names of the authors of the studies in, for example, fig 1 and 3. Also the panels miss the letters
Author response: We have improved the quality of the figures and included high quality tiff file as supplementary for the readers.
- Despite I found the study very sound, I think the discussion has quite some room for improvement. my two suggestions are 1) discuss further about potential mechanisms by which polymorphism in PD-1/PD-L1 can affect cancer susceptibility. e.g. do those changes result in change in the amino acid sequence of the protein? Is there any study that have correlated SNP with gene expression? are those codons predicted to have different translation efficiency? 2) I found intriguing that some of the mentioned studies have very distinct results from the others. I think this observation deserves a comment and an attempt of explaining the inconsistency (e.g. Haghshenas et al 2017 is a clear outlier in fig 1)
Author response: Thanks for your comments. We have added another table (Table 1) for the location and base-pair positions of single nucleotide polymorphisms (SNPs) of PD-1 and PD-L1 to answer the respected reviewer comment.
There is not any study which has correlated gene expression and SNP. We can’t comment about intriguing that some of the mentioned studies have very distinct results from the others as in some studies the limitation of their investigations have not been mentioned and it could be the reasons of these differences.

Round 2
Reviewer 1 Report
The authors have nicely and significantly improved their manuscript, having adequately addressed each of my comments. The figure quality has been improved. I have no further concerns with the manuscript.
Author Response
The authors team appreciate the respected reviewer positive feedback. We are all thankful for previous comments which have significantly improved the quality of our manuscript.
Reviewer 2 Report
The relevance of the study is made clearer in the introduction and discussion
Furthermore, figures are of much better quality and easier to understood.
Moreover, the subgroup analysis in table 4 concerning the relation between polymorphisms and different type of cancers is an added value for the manuscript.
After a minor English spelling and language style check, I would recommend publication of this manuscript.
Author Response
All authors appreciate the respected reviewer comment regarding spelling and grammar check of the revised manuscript. We have made careful correction of the manuscript and included them with track change.